# The comparative effectiveness of progressive relaxation training on pain characteristics, attack frequency, activity self-efficacy, and pain-related disability in women with episodic tension-type headache and migraine

**Aysenur Karakus** [12*¤a], **Esra Uzelpasaci** [3¤b], **Gokcen Akyurek** [2¤c]

**1** Health Sciences Faculty, Department of Occupational Therapy, Cankırı Karatekin University, Cankırı, Turkey, **2** Health Sciences Faculty, Department of Occupational Therapy, Hacettepe University, Ankara, Turkey, **3** Faculty of Gulhane Physiotherapy and Rehabilitation, University of Health Science, Ankara, Turkey

¤a Current address: Cankırı Karatekin University, Health Sciences Faculty, Department of Occupational Therapy, Cankırı, Turkey
¤b Current address: University of Health Science, Faculty of Gulhane Physiotherapy and Rehabilitation, Ankara, Turkey, e-mail address: uzelpasaciesra@gmail.com, Telephone number: +90 554 544 36 16.
¤c Current address: Hacettepe University, Health Sciences Faculty, Department of Occupational Therapy, Ankara, Turkey, e-mail address: gkcnakyrk@gmail.com, Telephone number: +90 0312 305 25 60/126.
* aykarakus_02@hotmail.com.tr

## Abstract

### Background and purpose

Episodic tension headache (TTH) and migraine, both categorized as primary headache types, account for 60–90% of headache complaints and are three times more common among young women. This study aimed to explore the comparative effectiveness of progressive relaxation training (PRT) on pain characteristics, attack frequency, activity self-efficacy, and pain-related disability in women with episodic TTH and migraine.

### Materials and methods

This study was registered at ClinicalTrials.gov (NCT06050382). This prospective study included women with episodic TTH (n=20) and migraine (n=20). The pain intensity, impact of headaches on life, activity self-efficacy, pain catastrophizing, and pain-related disability levels of both groups were measured using the Visual Analog Scale (VAS), Headache Impact Test (HIT-6), Occupational Self-Assessment Scale (OSAS), Pain Catastrophizing Scale (PCS), and World Health Organization Disability Assessment Schedule 2 (WHODAS-II) pre- and post-intervention, respectively. Both TTH and migraine groups received PRT twice a week for six weeks,

**Data availability statement:** Data cannot be shared publicly because of institutional privacy policies. Data are available from the Çankırı Karatekin University Ethics Committee. Institutional Data Access / Ethics Committee (contact via phone +90 376 218 95 00 and email etikkurul@karatekin.edu.tr) for researchers who meet the criteria for access to confidential data.

**Funding:** The author(s) received no specific funding for this work.

**Competing interests:** NO authors have competing interests

## Results

Within-group comparisons showed significant decreases in attack frequency, VAS, HIT-6, PCS, and WHODAS-II scores in both groups post-intervention (p<0.001). Also, both groups showed an increase in OSAS proficiency scores (p<0.001). The between-group comparison showed that the attack frequency, VAS, HIT-6, PCS, and WHODAS-II scores were lower in the migraine group than the TTH group. However, all sub-scores of the OSAS were higher in the migraine group (p<0.001).

## Conclusions

PRT showed positive effects on pain intensity, attack frequency, activity self-efficacy, and pain-related disability in both groups, more so in the TTH group.

---

## 1. Introduction

Headache is globally ranked as the third leading cause of disability [1]. Episodic Tension headache (TTH) and migraine, both categorized as primary headache types, account for 60–90% of headache complaints and are three times more common among young women compared to men [2,3]. Recurrent TTH and migraine attacks negatively impact women' quality of life by reducing their work capacity and participation in social and family activities. Thus, both TTH and migraine result in substantial social impairment, loss of work force, and economic burden [1–3].

Episodic Tension-type headache and migraine disrupt women' ability to engage in self-care, work/productivity and leisure activities due to symptoms including pain, exhaustion, impaired concentration, and dizziness [4,5]. Participation in purposeful activities is an indispensable part of health and well-being [6]. Activity self-efficacy includes individuals' sense of personal competence and satisfaction regarding the activities they engage in [6]. The sustainability of physical activity and productivity is linked to activity self-efficacy [6–8]. The primary complaint among women with TTH and migraine is that pain occurs at the ages of 25–55 years, typically considered the most productive and efficient period [7,8]. Furthermore, participation in daily life activities (self-care, school, and work), self-sufficiency, and self-management skills are negatively related due to direct or indirect physiological and morphological changes in the muscle groups that women frequently use in their daily activities [2,9–11]. In addition, constant preoccupation with pain and avoidance of physical activity may lead to a loss of function in women's roles and reduced participation in life [8–11]. Therefore, the unique and intricate nature of human roles implies that women diagnosed with headaches may encounter various restrictions in fulfilling their daily responsibilities and engaging in social activities, depending on their lifestyle and personal preferences as a result of pain [8–12].

In the literature, non-pharmacological intervention methods (physiotherapy interventions,biofedback lifestyle changes, cognitive behavioral therapy, and body awareness therapy) are recommended for women with TTH and migraine to reduce

pain, enable participation in daily life, and improve the quality of life [2,7,13]. One of these interventions is progressive relaxation training (PRT) [13,14]. PRT is defined as a method that induces relaxation in the whole body by voluntary and regular relaxation of major muscle groups of the human body [14]. It is particularly advantageous for muscle groups in the hands, arms, neck, shoulders, face, chest, abdomen, hips, feet and fingers, most commonly used in daily life activities [14]. PRT was reported to increase participation in activities of daily living (ADL) and quality of life in addition to pain management in individuals with various chronic pains [7,15–17]. However, to the best of our knowledge, there is no study comparing the advantageous effects of PRT, one of the optimal treatment methods applicable to women with TTH or migraine, across these two different headache groups. In addition, there were very few studies examining the effect of PRT on activity self-efficacy and social participation [7,11,18].

The aim of the present study was to investigate comparative effectiveness of PRT on pain characteristics, attack frequency, activity self-efficacy, and pain related disability among women experiencing two different headache types, TTH and migraine. The null hypothesis is that PRT has no differential effect on pain characteristics, attack frequency, activity self-efficacy, or pain-related disability between women with episodic TTH and those with migraine. The alternative hypothesis is that PRT is more effective in one group (TTH or migraine) compared to the other in improving pain characteristics, attack frequency, activity self-efficacy, and pain-related disability.

This study can enable clinicians dealing with women suffering from episodic TTH and migraine to typically make clinical evaluations more holistically. It can also shed light on planning more comprehensive intervention strategies to address the problems women encounter in their daily lives.

## 2. Materials and methods

### 2.1. Study design

This study is prospective, experimental between-group study with premeasurements and postmeasurements. In this study, TTH and migraines participated in PRT sessions twice a week for six weeks. The aim of this study was to evaluate and compare the outcomes of these two distinct groups after the training.

This was conducted in accordance with the rules of the Declaration of Helsinki. It was approved by the Çankırı Karatekin University Ethics Commission (Approval No: 3, Date: 08.11.2022) and was registered at ClinicalTrials.gov (NCT06050382). Written informed consent was obtained. The study was conducted between January 2023 and May 2023

### 2.2. Participants

Women were invited to the study through digital announcements, brochures and posters. Convenience sampling method was used to reach more women with TTH and migraine. The inclusion criteria were determined as follows; aged between 20 and 45 years diagnosed with episodic TTH and migraine by neurologist in accordance with the criteria of the International International Headache Society (ICHD-II) [2], and volunteering to participate in the study.

Women diagnosed with migraines within the last 3 months and women diagnosed with TTH within the last 3–6 months were included in the study [2].

The exclusion criteria were as follows; having a pathology involving the cervical region (such as disc herniation, radiculopathy, surgical history, tumor, cyst), having received any physical therapy for the cervical region within three months, being pregnant, being in menopause, not participating in 15% or more of the training, having a mental disorder (such as diagnosed with depression, using antidepressant medication), having a chronic, neurological, or rheumatic disorder, having sinusitis, and constantly using migraine prophylaxis medications. The study included 58 women between the ages of 20 and 45 years. Written and verbal consent was obtained by providing information to the women (n=58) about the nature and purpose of the study, potential risks and benefits, and their rights to withdraw at any time without any consequences. However, only 40 women completed the study. The details of included and excluded women into the study are provided as a flowchart (Fig. 1).

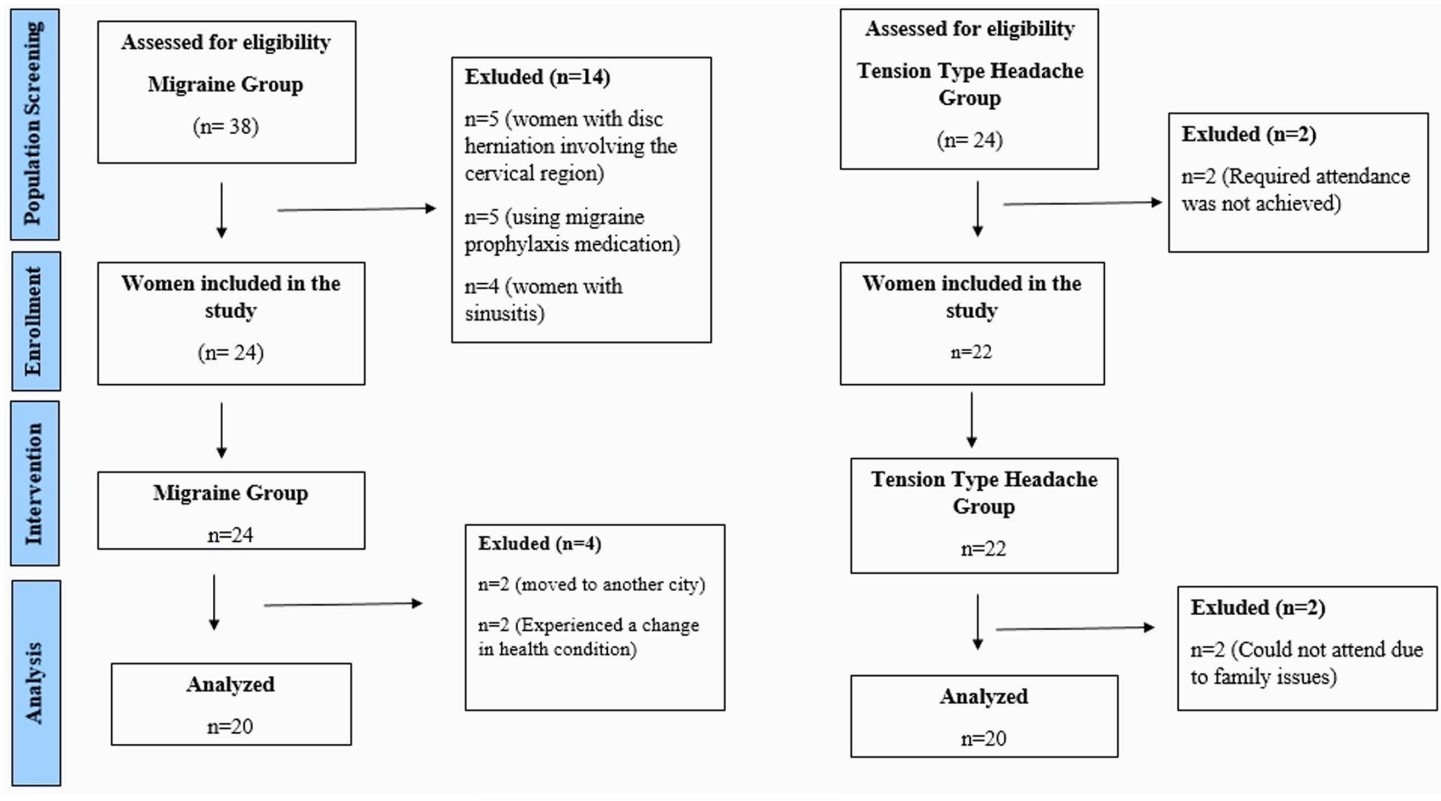

**Fig 1. Flow chart.**

## 2.3. Interventions

PRT, defined by Jacobson as contracting and relaxing 16 muscle groups [14], was performed two days a week for a total of 12 sessions for a period of six weeks in both TTH and migraine groups. The sessions were performed in a quiet, comfortable, and dimly lit room for 30 minutes. Women were asked to stop eating and wear comfortable clothes at least two hours before the training session. They were first taught deep diaphragmatic breathing. Following diaphragmatic breathing, PRT was initiated in supine position and was progressed by contracting the muscles from distal to proximal (starting from the feet to the lower extremities, trunk, upper extremities, and facial muscles, respectively) for 10–20 seconds and relaxing them for 30–40 seconds. During the relaxation period, the participants continued deep and slow breathing while moving to the upper part of the body. Each region was exercised with one repetition for a total of 16 regions/muscle groups. Safety measures were implemented during the PRT to ensure the well-being of the participants. Women were instructed to perform the trainings in a safe environment and were advised to stop immediately and report any pain or discomfort they experienced. PRT sessions were conducted in a controlled environment with soundproofing, ergonomic seating, privacy measures, accessibility features, emergency preparedness, optimal lighting, and temperature and humidity control to maximize safety and comfort for all participants. Progressive relaxation training was separately provided to both groups. The following evaluations were made for both groups before(baseline) and after the training (6 weeks). The monitoring process was conducted through weekly meetings and regular phone calls with the participants. The same physiotherapist practitioner (A.K.) did evaluations and progressive relaxation training in Cankırı Karatekin University, Health Sciences Faculty, Department of Occupational Therapy.

## 2.4. Outcomes

Demographic variables such as age, education, working and marital status, and clinical variables including height in centimeters, weight in kilograms, and body mass index in kg/m² of the women were obtained. A form was created for these informations.

**2.4.1. Primary outcome. 2.4.1.1. Visual Analog Scale (VAS):** The VAS was used to assess the pain intensity. It consists of 10 cm long horizontal lines, where 0 defines 'no pain' and 10 represents 'worst imaginable pain'. Women were asked to mark the intensity of their headaches on the horizontal lines and the results were recorded in centimeters [19]. The frequency and duration of attacks in women were evaluated self-reportedly, and the frequency and duration of attacks in the last month were questioned.

**2.4.2. Secondary outcomes. 2.4.2.1. Headache Impact Test (HIT-6):** HIT-6 is a validated questionnaire consisting of six questions, including the frequency of headache, severity of pain, fatigue and mood changes due to headache, degree of limitation in ADL, and social environment due to pain. Each question has the following response options: never (6 points), rare (8 points), sometimes (10 points), very often (11 points), and always (13 points) (20). Headache impact on this scale ranges from 36 (no headache) to 78 (very severe headache). Gadnek et al. (2003) developed HIT-6 and Dikmen et al (2020) conducted Turkish validity and reliability study. Its Cronbach's alpha validity coefficients range between 0.753 and 0.864 [20,21].

**2.4.2.2. Occupational Self-Assessment Scale (OSAS):** OSA is a self-reported and client-centered evaluation tool. It is based on the Model of Human Occupation, which assesses the degree of self-awareness and importance of the occupation by measuring the client's ability and value in performing routine occupation [22,23]. A total of four points are obtainable from each of the 21 questions. Questions related to ability are assessed as: "very difficult", "somewhat difficult", "can do well", and "can do very well". Questions related to value are assessed as: "not at all important", "somewhat important", "important", and "very important". These scores are transformed into scores from 0 to 100 using the method in the scale's application guide, resulting in two separate total scores, one for ability and one for value. Higher scores in OSA reflect higher self-recognition levels regarding the ability and value of one's occupational performance. Baron et al. (2009) originally developed the scale and Pekçetin et al. (2020) conducted Turkish validity and reliability study. The Cronbach's alpha coefficients for all subdimensions ranged between 0.95 and 0.96 [22,23].

**2.4.2.3. World Health Organization Disability Assessment Schedule 2, WHO-DAS II:** The 36-item WHO-DAS II, which consists of six subdimensions (understanding and communicating, getting around self-care, getting along with people, life activities-household and work, and participation in society), assesses social participation and functioning [24,25]. Women are asked to rate the level of difficulty they experienced during the related activity in the last month using a scale of 1–5, with responses ranging from none, mild, moderate, very much, extremely/not at all. Raw scores are transformed into scales ranging from 0 to 100. Low values indicate good performance. Ustun et al (2004) developed WHO-DAS II and Ulug et al. (2001) conducted Turkish validity and reliability study. The Cronbach's alpha coefficients for all subdimensions ranged between 0.60 and 0.90 [24,25].

**2.4.2.4. Pain Catastrophizing Scale (PCS):** PCS consists of 13 items, with each item evaluated between 0 and 4 points and a total score of 0–52 [26,27]. High scores indicate a high level of catastrophizing. Sullivan et al. (1995) developed the PCS and Süren et al. (2014) conducted Turkish validity and reliability study. The Cronbach's alpha values of the PCS ranges between 0.95 and 0.96 [26,27].

## 2.5. Statistical analysis

The G* Power software (G* Power, Version 3.1.9.7 Franz Faul, Universität Kiel, Germany) was used to determine the sample size. The preliminary hypothesis was defined as difference in VAS scores over time (pre-test and post-test) and between groups. Accordingly, it was assumed that the time and group effects would have a moderate effect size in our

study. With a Type I error rate of α = 0.05 α = 0.05 (95% confidence level) and a desired power of 1 − β = 0.80 1 − β = 0.80, the required sample size for statistical analyses was calculated as 20 (n=20).

Post-hoc power analysis was conducted after the study to evaluate whether the sample size used was sufficient. The post-hoc power analysis revealed that the power of the study was 98%. This indicates that the sample size used in the study was statistically sufficient.

In this study, to minimize potential bias, the statistician responsible for analyzing the data was blinded to the group allocations throughout the study.

IBM SPSS 26.0 (Statistical Package for the Social Sciences, IBM Corp., Armonk, NY, USA, 2019) was used for statistical analysis. Descriptive statistics included mean ± standard deviation for numerical variables, and number (n) and percent (%) values for categorical variables. The dependent sample t-test was used to examine the differences in terms of measurements over time, while the chi-square and independent sample t-tests were used to examine the differences between groups. In addition, Cohen's d value was provided for effect sizes. The significance level was set at $p < 0.05$.

## 3. Results

Table 1 provides demographic data of the participants categorized into groups. There were no differences between the group in terms of baseline physical characteristics except for working status ($p < 0.05$).

Before treatment, the migraine group showed higher numbers of monthly and hourly attacks, VAS, and HIT-6 scores compared to the TTH group ($p < 0.001$; d = −0.701; d = −4.220; d = −1.446; -2.163, respectively). After treatment, within-group comparisons showed a decrease in the number of monthly and hourly attacks, VAS and HIT-6 scores in both the migraine group ($p < 0.001$; d = −1.018; d = −0.695; d = −0.872; d = −1.487, respectively) and TTH group ($p < 0.001$; d = −2.181; d = −1.540; d = −6.367; d = −3.330, respectively). However, between group comparisons showed that the number of monthly and hourly attacks, VAS, and HIT-6 scores were higher in the migraine group than TTH group ($p < 0.05$) (Table 2).

**Table 1. Demographic and physical characteristics of the women.**

| | | Migraine Group | | TTH Group | | Test | p |
|---|---|---|---|---|---|---|---|
| | | (n = 20) | | (n = 20) | | | |
| Age (year) | X ± SD, min-max | 26.40 ± 3.84 (21–34) | | 26.45 ± 3.78 (20–33) | | −0.041[a] | 0.967 |
| BMI (kg/m$^2$) | X ± SD, min-max | 22.48±1.34 (20.20–26.56) | | 22.03 ± 0.93 (20.55–23.51) | | 1.208[a] | 0.234 |
| Duration of education (year) | X ± SD, min-max | 16.00 ± 1.30 (12–18) | | 16.30 ± 1.34 (16–22) | | −0.719[a] | 0.477 |
| | | n | % | n | % | | |
| Education level, n (%) | High school | 5 | 25.0 | 0 | 0.0 | – | – |
| | Undergraduate | 1 | 5.0 | 0 | 0.0 | | |
| | University | 11 | 55.0 | 19 | 95.0 | | |
| | Master's degree Graduate | 3 | 15.0 | 0 | 0.0 | | |
| | PhD graduate | 0 | 0.0 | 1 | 5.0 | | |
| Working status, n (%) | Yes | 9 | 45.0 | 2 | 10.0 | 6.144[b] | 0.013* |
| | No | 11 | 55.0 | 18 | 90.0 | | |
| Marital status, n (%) | Married | 3 | 15.0 | 2 | 10.0 | 0.229[b] | 1.000 |
| | Single | 17 | 85.0 | 18 | 90.0 | | |

*: $p < 0.05$, X: Mean, SD: Standard deviation

[a]: Independent sample t test,

[b]: Chi-square test, n: frequency, %: Percentage, BMI: Body mass index, kg: Kilograms, m: Meter, TTH: Tension type headache.

The ability and value subdimension scores of OSA were lower in the migraine group than in the TTH group before treatment ($p < 0.05$; d = 1.062; d = 1.215, respectively). Within-group comparisons showed that both the migraine group ($p < 0.001$; d = 1.484) and TTH group ($p < 0.001$; d = 2.444) showed improvement by increasing their scores in the OSA-ability subdimension. However, OSA-value scores were found to be similar ($p > 0.05$). The results of between-group comparisons showed that the ability and value scores of OSA were lower in the migraine group compared to the TTH group after treatment ($p < 0.001$) (Table 2).

Considering the groups in terms of PCS, rumination scores were similar between the groups before treatment ($p > 0.05$; d = –0.369). In addition, the total scores, as well as magnification and helplessness subdimension of the PCS were higher in the migraine group compared to the TTH group before treatment ($p < 0.05$; d = −0.690; d = −1.720; d = −2.260, respectively) (Table 2).

Within-group comparisons showed that there was a decrease in rumination, helplessness and total PCS scores except for the magnification subdimension in the migraine group after treatment. Additionally, there was a significant decrease in all subdimensions and total scores of PCS in the TTH group in the within-group comparison ($p < 0.001$). In the post-treatment comparison between the groups, the PCS subdimensions and total scores were higher in the migraine group than in the TTH group ($p < 0.001$; d = −1.532; d = −1.374; d = −1.927; d = -−.000, respectively) (Table 2).

The work activities and life activities total scores were similar between the groups before treatment ($p>0.05$: d=-0.139;d=-0.591, respectively). In addition, the total scores of WHODAS-II subdimensions, except for the getting along with people subdimension, were higher in the migraine group compared to the TTH group before treatment ($p<0.001$). Both TTH and migraine groups significantly decreased and improved in terms of all subdimension scores of WHODAS except for the frequency of challenges experienced by women (H1) in within-group comparisons after treatment ($p<0.05$). The migraine group had higher disability in terms of WHODAS subdimensions and total scores compared to the TTH group in the between-group comparisons after treatment ($p<0.05$)(Table 3).

In post-treatment comparisons, the TTH group had a lower number of attacks (monthly) compared to the migraine group (t=2.230; p=0.032). Pain intensity (VAS) was lower in the TTH group than in the migraine group (t=6.036; p<0.001). Similarly, HIT-6 total scores, reflecting the impact of headaches, were lower in the TTH group compared to the migraine group (t=6.463; p<0.001). Regarding WHODAS-II subdomains, the TTH group demonstrated lower scores in "understanding and communicating" (t=3.060; p=0.004), "getting around" (t=2.484; p=0.018), "self-care" (t=4.454; p<0.001), and "getting along with people" (t=5.595; p<0.001). Additionally, the total scores for life activities were lower in the TTH group than in the migraine group (t=5.661; p<0.001). However, no significant differences were found between the groups in OSAS-Value (t=0.882; p=0.384), PCS-Helplessness (t=1.377; p=0.177), H1 (t=1.560; p=0.127), H2 (t=1.209; p=0.234), and H3 (t=1.505; p=0.141) (Table 4).

## 4. Discussion

This study evaluated the comprative effectiveness of PRT in two different types of headaches, TTH and migraine, by assessing its effects on pain characteristics, attack frequency, activity self-efficacy, and pain-related disability in women with episodic TTH and migraine. Both within-group and between-group differences were analyzed to determine whether one group experienced more significant benefits from PRT compared to the other. The results revealed that PRT was effective in reducing pain intensity and attack frequency, while also positively impacting activity self-efficacy and pain-related disability in women with both TTH and migraine. Notably, between-group comparisons demonstrated that PRT was more effective in the TTH group.

This study is significant for health professionals treating women with episodic tension-type headache and migraine. Treatment methods designed to address headaches in women may incorporate PRT either as a standalone approach or as a supplementary method.

The study sample group comprised young adult women due to the higher prevalence of tension-type headache (TTH) and migraine in this demographic, as reported in the literature [1–3]. The literature also reports that migraine causes more

**Table 2. Comparison of the number of attacks, pain intensity, OSA and PCS within-group and between groups at baseline and after six weeks.**

| Within-group comparison | | Migraine Group | | TTH Group | | t[1] | p | Between-group comparison Cohen's d |
|---|---|---|---|---|---|---|---|---|
| | | X±SD | Median (Min-Max) | X±SD | Median (Min-Max) | | | |
| Number of attacks (months) | Baseline | 4.40±1.67 | 4 (2-8) | 3.45±0.94 | 4 (2-5) | 0.022 | 0.001** | -0.701 |
| | After six weeks | 3.55±1.36 | 3.5 (1-4) | 2.00±1.34 | 2.5 (0-4) | 3.639 | 0.001* | -1.148 |
| | t[2]/p Cohen's d | 4.677/0.001** -1.018 | | 7.310/0.001** -2.181 | | | | |
| Number of attacks (hours) | Baseline | 56.10±16.55 | 51 (24-72) | 6.25±2.27 | 6 (3-12) | 13.346 | 0.001** | -4.220 |
| | After six weeks | 47.40±11.33 | 48 (24-72) | 3.90±2.86 | 4.5 (0-8) | 16.641 | 0.001** | -5.265 |
| | t[2]/p Cohen's d | 3.356/0.003* -0.695 | | 5.792/0.001** -1.540 | | | | |
| VAS (cm) | Baseline | 7.57±0.98 | 7.4 (6-10) | 6.39±0.61 | 6.4 (5-7.4) | 4.579 | 0.001** | -1.446 |
| | After six weeks | 6.76±0.65 | 7 (5.5-7.8) | 3.04±2.01 | 3.6 (0-5.4) | 7.898 | 0.001** | -2.490 |
| | t[2]/p Cohen's d | 4.380/0.001** -0.872 | | 8.863/0.001** -6.367 | | | | |
| HIT-6 total score | Baseline | 70.30±2.83 | 70 (66-78) | 61.75±4.82 | 61.5 (52-70) | 6.839 | 0.001** | -2.163 |
| | After six weeks | 66.85±2.43 | 66.5 (62-70) | 50.00±7.70 | 49 (36-64) | 9.333 | 0.001** | -2.951 |
| | t[2]/p Cohen's d | 7.052/0.001** -1.487 | | 9.896/0.001** -3.330 | | | | |
| OSA | | | | | | | | |
| OSA-Ability | Baseline | 32.00±5.42 | 32.5 (21-44) | 37.70±5.31 | 39.5 (28-44) | -3.359 | 0.002* | 1.062 |
| | After six weeks | 43.35±2.30 | 43 (41-51) | 52.70±9.03 | 51.5 (36-64) | -4.489 | 0.001** | 1.419 |
| | t[2]/p Cohen's d | 8.635/0.001** 1.484 | | -7.601/0.001** 2.444 | | | | |
| OSA-Value | Baseline | 50.30±5.09 | 50 (42-61) | 55.40±3.05 | 55.5 (51-61) | -3.844 | 0.001** | 1.215 |
| | After six weeks | 51.00±4.94 | 50.5 (43-61) | 55.50±2.9 | 1 56 (52-61) | -3.509 | 0.001* | 1.110 |
| | t[2]/p Cohen's d | -1.129/0.273 0.249 | | -0.357/0.725 0.079 | | | | |
| PCS | | | | | | | | |
| Rumination | Baseline | 6.85±1.18 | 7 (5-8) | 6.30±1.75 | 7 (3-8) | 1.165 | 0.252 | -0.369 |
| | After six weeks | 5.85±1.04 | 6 (4-8) | 3.35±2.06 | 4 (0-6) | 4.847 | 0.001** | -1.532 |
| | t[2]/p Cohen's d | 4.595/0.001** -0.976 | | 6.472/0.001** -1.589 | | | | |
| Magnification | Baseline | 4.60±2.48 | 5 (1-9) | 3.20±1.44 | 3 (0-6) | 2.185 | 0.037* | -0.690 |
| | After six weeks | 4.70±2.18 | 5 (1-8) | 2.15±1.46 | 2,5 (0-5) | 4.347 | 0.001** | -1.374 |
| | t[2]/p Cohen's d | -0.317/0.755 0.068 | | 2.868/0.010* -0.645 | | | | |
| Helplessness | Baseline | 7.45±1.76 | 8 (4-11) | 4.70±1.42 | 4 (2-7) | 5.439 | 0.001** | -1.720 |
| | After six weeks | 6.70±1.75 | 7 (3-9) | 3.20±1.88 | 4 (0-6) | 6.093 | 0.001** | -1.927 |
| | t[2]/p Cohen's d | 2.595/0.018* -0.579 | | 3.249/0.004* -0.857 | | | | |

*(Continued)*

**Table 2.** (Continued)

| Within-group comparison | | Migraine Group | | TTH Group | | t[1] | p | Between-group comparison Cohen's d |
|---|---|---|---|---|---|---|---|---|
| | | X±SD | Median (Min-Max) | X±SD | Median (Min-Max) | | | |
| Total score | Baseline | 35.05±4.24 | 34.5 (26-43) | 25.70±4.03 | 26.5 (20-33) | 7.154 | 0.001** | -2.260 |
| | After six weeks | 30.75±3.27 | 30 (23-38) | 16.45±9.57 | 19 (0-31) | 6.321 | 0.001** | -2.000 |
| | t[2]/p Cohen's d | 5.549/0.001** -1.134 | | 4.368/0.001** -1.857 | | | | |

*: p<0.05,

**p<0.001, X: Mean, SD: Standard deviation, Min: minimum Max: maximum, t[1]: Independent sample t test, t[2]: Dependent sample t test, d: Cohen's d value, cm: Centimeters, TTH: Tension type headache

pain and pain-related disability than TTH. In this study, the migraine group had higher pain intensity, attack frequency, and pain-related disability levels before treatment compared to the TTH group. These results align with studies in the literature reporting that migraine-related headaches are more severe than TTH and negatively affect participation in ADL and quality of life [3–5]. In addition, episodic migraine and TTH were included in the study since the duration and severity of attacks in episodic migraine and TTH are likely to be similar compared to chronic migraine and tension-type headache [5].

The results of this study showed that PRT significantly reduced both the frequency and severity of headaches in both migraine and TTH. However, there were differences in the literature regarding the effect of PRT on migraine and tension-type headache [6,15,28,29]. Fichtel and Larsson (2001) reported that the frequency and severity of headache in adolescents (n=18 females, n=18 males) decreased in the migraine group but did not change in TTH [15]. Kumar et al. (2014) compared the efficacy of transcutaneous electrical stimulation (TENS) and PRT in women and men with TTH. They reported that while PRT reduced pain and stress, TENS significantly reduced stress but was not effective in reducing pain [30]. Dittrich et al. (2008) reported that relaxation training provided in combination with aerobic exercise decreased pain intensity in men and women with migraine compared to the control group. However, there was no difference between the groups in terms of psychological variables [31]. The present study found a decrease in pain intensity, frequency of attacks on a monthly and hourly basis in both the TTH and migraine groups after treatment, consistent with existing literature. The reduction in pain intensity, monthly and hourly number of attacks was higher in the TTH group compared to the migraine group. The improved results of PRT in the TTH group, as compared to the migraine group, may be related to differences in the pathophysiology of TTH and migraine [32,33]. Migraine is often associated with neurovascular mechanisms and heightened central sensitization, whereas TTH is predominantly linked to muscular tension and peripheral pain mechanisms. These differences may influence the effectiveness of PRT, which targets muscle tension and promotes overall relaxation [4,32,33]. Also, the pathophysiology of migraine can be related to ovarian hormones and changes in estrogen levels, including hormonal changes and menstrual cycle variations in women. These can increase the sensitivity of dopamine receptors and change the serotonergic transmission mechanism and vascular structure [32]. Considering the pathophysiology of TTH, pain may arise due to irregular metabolism in the periphery muscles, which can lead to an inflammatory response, decreased blood flow, increased muscle activity, and muscle atrophy [33]. PRT was reported to normalize blood pressure, decrease oxygen consumption, respiration, heart rate, and muscle tension [14–16]. It is hypothesized that relaxation is effective in reducing pain by (a) reducing the level of chemicals such as lactic acid that trigger pain by decreasing tissue oxygen demand, (b) reducing skeletal muscle tension and anxiety that may exacerbate pain, and (c) stimulating endorphin secretion [14,33]. Thus, PRT is likely to be more effective in reducing pain intensity and attack frequency in TTH compared to the migraine group [14,32,33].

**Table 3. Comparison of WHODAS-II scores at baseline and after six weeks.**

| Within-group comparison | | Migraine Group | | TTH Group | | t[1] | p | Between-group comparison Cohen's d |
|---|---|---|---|---|---|---|---|---|
| | | X±SD | Min-Max | X±SD | Min-Max | | | |
| Understanding and communicating | Baseline | 19.90±3.93 | 18.5 (14-30) | 15.40±2.56 | 15.5 (10-21) | 4.288 | 0.001** | -1.357 |
| | After six weeks | 17.75±2.92 | 17.5 (12-24) | 10.74±3.77 | 11 (6-18) | 6.517 | 0.000* | -2.079 |
| | t[2]/p Cohen's d | 3.640/0.002* -0.760 | | 7.398/0.001** -2.185 | | | | |
| Getting around | Baseline | 19.95±3.35 | 20 (10-25) | 13.75±1.71 | 14 (11-17) | 7.373 | 0.001** | -2.331 |
| | After six weeks | 16.85±2.52 | 17 (10-20) | 8.84±2.69 | 1 (5-14) | 9.596 | 0.001** | -3.073 |
| | t[2]/p Cohen's d | 6.0490001** -1.257 | | 8.427/0.001** -2.520 | | | | |
| Self-care | Baseline | 12.50±2.31 | 12 (9-18) | 10.10±1.33 | 10 (8-13) | 4.030 | 0.001** | -1.273 |
| | After six weeks | 11.25±2.51 | 11 (8-20) | 6.21±1.62 | 6 (4-9) | 7.406 | 0.001** | -2.386 |
| | t[2]/p Cohen's d | 3.455/0.003* -2.755 | | 8.273/0.001** -2.139 | | | | |
| Getting along with people | Baseline | 11.80±2.61 | 12 (5-16) | 13.70±1.45 | 13.5 (11-16) | -2.846 | 0.007* | 0.900 |
| | After six weeks | 10.00±2.43 | 10.5 (5-14) | 8.21±2.18 | 9 (5-11) | 2.420 | 0.021* | -0.775 |
| | t[2]/p Cohen's d | 4.904/0.001** -1.064 | | 9.801/0.001** -2.885 | | | | |
| Life activities – excluding work | Baseline | 16.85±2.48 | 17(12-20) | 14.05±2.11 | 14 (10-18) | 3.845 | 0.001** | -1.216 |
| | After six weeks | 13.65±3.10 | 14 (8-18) | 7.68±3.16 | 8 (1-12) | 5.948 | 0.001** | -1.907 |
| | t[2]/p Cohen's d | 3.850/0.001** -0.836 | | 10.419/0.001** -3.204 | | | | |
| Work activities | Baseline | 14.45±4.57 | 16 (4-19) | 13.95±2.26 | 14 (11-18) | 0.439 | 0.663 | -0.139 |
| | After six weeks | 12.15±4.42 | 12 (0-17) | 8.32±2.79 | 8 (4-12) | 3.222 | 0.003* | -1.036 |
| | t2/p Cohen's d | 4.524/0.001** -0.999 | | 10.065/0.001** -2.535 | | | | |
| Life activities | Baseline | 30.55±4.47 | 31.5 (20-36) | 28.00± 4.15 | 27.5 (21-36) | 1.870 | 0.069 | -0.591 |
| | After six weeks | 26.60±7.10 | 28 (4-34) | 15.42±6.30 | 16 (2-24) | 5.192 | 0.001** | -1.666 |
| | t[2]/p Cohen's d | 4.031/0.001* -1.419 | | 10.495/0.001** -3.213 | | | | |
| Participation | Baseline | 23.50±2.80 | 23 (19-30) | 20.80±2.44 | 21 (16-25) | 3.250 | 0.002* | -1.028 |
| | After six weeks | 20.45±2.33 | 21 (15-24) | 14.74±4.04 | 16 (8-21) | 5.375 | 0.001** | -1.731 |
| | t[2]/p Cohen's d | 5.498/0.001* -1.142 | | 7.138/0.001** -2.296 | | | | |

*(Continued)*

**Table 3.** (Continued)

| Within-group comparison | | Migraine Group | | TTH Group | | t[1] | p | Between-group comparison Cohen's d |
|---|---|---|---|---|---|---|---|---|
| | | X±SD | Min-Max | X±SD | Min-Max | | | |
| WHODAS-II total score | Baseline | 118.20± 119 | (91-129) 8,64 | 101.75± 102 | (88-122) 9.99 | 5.785 | 0.001** | -1.830 |
| | After six weeks | 102.9±11,30 | 105(71-121) | 65.05+17.44 | 68(36-97) | 8.698 | 0.001** | -2.772 |
| | t[2]/p Cohen's d | 3.447/0.001** -2.748 | | 10.763/0.001** -3.986 | | | | |
| H1 | Baseline | 9.55±2.35 | 9 (6-15) | 5.85±2.08 | 5.5(2-9) | 5.267 | 0.001** | -1.667 |
| | After six weeks | 8.55±2.35 | 8 (5-16) | 3.84±2.3 | 4 5(0-7) | 6.267 | 0.001** | -2.009 |
| | t2/p Cohen's d | 1.541/0.140 -0.345 | | 5.407/0.001** -1.184 | | | | |
| H2 | Baseline | 9.55±2.35 | 9(6-15) | 5.85±2.08 | 5.5(2-9) | 5.267 | 0.001** | -1.667 |
| | After six weeks | 7.95±1.73 | 8(5-10) | 3.84±2.34 | 5(0-7) | 6.255 | 0.001** | -1.997 |
| | t2/p Cohen's d | ,5.287/0.001** -1.141 | | 5.407/0.001** -1.184 | | | | |
| H3 | Baseline | 9.55±2.35 | 9(6-15) | 5.85±2.08 | 5.5(2-9) | 5.267 | 0.001** | -1.667 |
| | After six weeks | 8.10±1.59 | 8(5-10) | 3.84±2.34 | 5(0-7) | 6.683 | 0.001** | -2.130 |
| | t2/p Cohen's d | 4.781/0.000* -1.064 | | 5.407/0.001** -1.184 | | | | |

*: p<0.05,

**p=0.000, X: Mean, SD: Standard deviation, Min: minimum Max: maximum, t1: Independent sample t test, t2: Dependent sample t test, d: Cohen's d value, WHODAS II: World Health Organization Disability Assessment Schedule 2 (WHODAS-II), H1: How frequently did women experience these difficulties, H2: How frequently did the difficulties prevent them carrying out their daily activities or work, H3: How frequently did the difficulties cause them to reduce their daily activities

Studies reported that headache is characterized by uncertainty in the timing, duration and causes of attacks, resulting in fatigue, altered emotional state and inability to participate in activities, resulting in disability and impairment of individuals' ability to fulfill their roles in life (work, school and social life) [9,11,13]. Kwekkeboom and Gretarsdottir (2006) reported that PRT administered along with deep breathing exercises for 12 weeks reduced pain intensity and disability due to headache in a randomized controlled study conducted with individuals with TTH [34]. Minen et al.(2020) reported that telephone-assisted PRT reduced the level of headache-related disability [35]. Therefore, PRT decreased the severity of pain, which may have a positive impact on ADL and quality of life [34–37]. In accordance with the literature, this study found that PRT decreased the impact of headaches in women with TTH and migraine. PRT may have alleviated different obstacles associated with headaches or facilitated coping by inducing muscular relaxation [34–37].

Catastrophizing pain causes the progression of pain into a chronic condition, restricting the individual's daily life activities and having negative effects on the their self-efficacy [38–41]. In this study, PRT decreased in all subdimensions except for the magnification in the migraine group and decreased in all subdimensions of PCS in the TTH group. PRT typically reduces negative thought patterns (magnification and catastrophizing) and behaviors (avoidance or inactivity due to pain) that increase chronic pain by stimulating the parasympathetic nervous system and increasing the release of endorphin mediators [14,16]. This mechanism may positively affect the thought patterns associated with catastrophizing pain in TTH and migraine [38–41]. PRT may have altered the pain catastrophizing pattern by breaking the negative cognitive patterns associated with pain, or by enhancing the functioning of the damaged muscle groups [38–41]. Also, PRT increased the ability to perceive activity self-efficacy in both groups in the current study. However, the perception of value related to activity did not change. The literature suggests that pain coping techniques like PRT positively affect activity

**Table 4. Comparison of changes between migraine and tension-type headache groups (Δ Scores).**

| Delta (Δ) | Migraine | | GTBA | | t | p |
|---|---|---|---|---|---|---|
| | X±SD | Median (min-max) | X±SD | Med (min-max) | | |
| Number of attacks (months) | -0,85±0,81 | -1 (-2-0) | -1,45±0,89 | -1 (-3-0) | 2,230 | 0,032 |
| Number of attacks (hours) | -8,7±11,59 | 0 (-24-0) | -2,35±1,81 | -2 (-6-2) | -2,420 | 0,025 |
| VAS (cm) | -0,81±0,83 | -0,7 (-3-0,5) | -3,35±1,69 | -2,9 (-6,5--1) | 6,036 | 0,001** |
| HIT-6 total score | -3,45±2,19 | -3,5 (-8-0) | -11,75±5,31 | -11 (-21--2) | 6,463 | 0,001** |
| **OSAS** | | | | | | |
| OSAS-Ability | 11,35±5,88 | 10 (-1-20) | 15±8,83 | 15 (0-31) | -1,539 | 0,132 |
| OSAS-Value | 0,7±2,77 | 0 (-4-10) | 0,1±1,25 | 0 (-3-4) | 0,882 | 0,384 |
| **PCS** | | | | | | |
| Rumination | -1±0,97 | -1 (-3-1) | -2,95±2,04 | -2,5 (-8-0) | 3,861 | 0,001* |
| Magnification | 0,1±1,41 | 0 (-2-4) | -1,05±1,64 | 0 (-6-0) | 2,380 | 0,022 |
| Helplessness | -0,75±1,29 | -1 (-3-3) | -1,5±2,06 | -1 (-7-1) | 1,377 | 0,177 |
| Total Score | -4,3±3,47 | -4 (-12-3) | -9,25±9,47 | -6 (-31-3) | 2,195 | 0,038 |
| **WHODAS-II** | | | | | | |
| Understanding and communicating | -2,15±2,64 | -2 (-8-2) | -4,84±2,85 | -5 (-10-0) | 3,060 | 0,004 |
| Getting around | -3,1±2,29 | -2,5 (-10-0) | -5,05±2,61 | -4 (-10--1) | 2,484 | 0,018 |
| Self-care | -1,25±1,62 | -1 (-4-2) | -3,89±2,05 | -3 (-7--1) | 4,454 | 0,001** |
| Getting along with people | -1,8±1,64 | -1 (-5-1) | -5,53±2,46 | -6 (-9--1) | 5,595 | 0,001** |
| Life activities – excluding work | -3,2±3,91 | -1,5 (-12-5) | -6,37±2,77 | -6 (-11--3) | 2,902 | 0,006 |
| Work activities | -2,3±2,27 | -2 (-8-2) | -5,79±2,51 | -5 (-12--3) | 4,557 | 0,001** |
| Life activities | -3,95±4,38 | -3,5 (-16-3) | -12,74±5,29 | -13 (-23--6) | 5,661 | 0,001** |
| Participation | -3,05±2,48 | -2,5 (-8-1) | -6,16±3,76 | -6 (-16-0) | 3,062 | 0,004 |
| WHODAS-Total | -15,30±7,44 | -14(-32/-5) | -38,32±14,09 | -35(-73/-23) | 6,240 | 0,001** |
| H1 | -1±2,9 | -2 (-5-10) | -2,21±1,78 | -2 (-6-0) | 1,560 | 0,127 |
| H2 | -1,6±1,35 | -2 (-5-2) | -2,21±1,78 | -2 (-6-0) | 1,209 | 0,234 |
| H3 | -1,45±1,36 | -1,5 (-5-2) | -2,21±1,78 | -2 (-6-0) | 1,505 | 0,141 |

*: p<0.05,

**p=0.000, X: Mean, SD: Standard deviation, Min: minimum Max: maximum, Delta (Δ): Difference between after six weeks scores (post-test) and baseline (pre-test), t: Independent Samples t-test, H1: How frequently did women experience these difficulties, H2: How frequently did the difficulties prevent them carrying out their daily activities or work, H3: How frequently did the difficulties cause them to reduce their daily activities.

and self-efficacy perception [9,10,42]. Leroux et al. (2018) showed that teaching coping skills was positively related to enhanced self-efficacy in individuals struggling with headaches [43]. Based on the results of the study, it becomes possible to increase individuals' sense of efficacy related to activity by encouraging increased activities rather than passive avoidance in activities and ensuring participation in ADL (i.e., self-management of pain) [9,10,42,43]. Therefore, PRT can increase activity participation by changing women's perspective on pain, having a positive impact on activity self-efficacy [9,10,41–44].

Studies have reported that 31% to 51% of individuals diagnosed with TTH and migraine experience more absenteeism from work and school, working hours, decreased work participation and productivity, and deterioration in relationships with friends, colleagues and family [3,4,9]. Another hypothesis is that the occurrence of spasm in the head and neck muscles with the effect of postural, social and environmental conditions triggers headache. Therefore, individuals avoid activities

and social environment that may cause spasm and find it challenging to fulfill their roles [7,10,11,15]. In the current study, PRT significantly reduced pain related disability in women with both types of headaches. The literature has shown that PRT positively affect daily life participation by positively supporting homeostasis, emotional balance, and coping skills with stress. Furthermore, this helps individuals to fulfill their roles by regulating their social relations [18,45,46]. The findings in our study are consistent with the literature.

There were some strengths and limitation in this study. One strength was the comparative demonstration of the effectiveness of PRT intervention in two different headache groups, migraine and TTH. In the study, women with higher education levels were present in both groups. 95.0% of women in the Migraine group were university graduates, while in the TTH group, 55.0% were university graduates. Education reflects an individual's intellectual abilities and may also be associated with different health behaviors [47]. Both women with TTH and migraine were a high level of education. It may have been advantageous in enabling patients to better understand and apply the PRT and thus reveal the true effect of the PRT in both groups [47].

Another strength of the study was that the number of participants was adequate for an intervention study. The study's limitations include the absence of blinding, as the researchers conducting the intervention and the evaluation were the same. In addition, women with episodic migraine and TTH were included in the study. Therefore, it may be better to include women with chronic migraine and TTH in future studies. In addition, although this study has a valuable aspect of being conducted on women, it has a limitation in terms of generalizability.

In conclusion, PRT decreased pain intensity and attack frequency, while increasing activity self-efficacy perception and social participation in individuals diagnosed with migraine and TTH. Demonstrating the effectiveness of PRT in headache is clinically crucial as it is a non-pharmacological, low side effect, and easily applicable method. This study plays a critical role as it demonstrates the application of PRT, which is complementary and often used alone, in different types of headaches. It enables health professionals to increase awareness of the intervention for women with headaches and the quality of person-centred intervention. Future studies should consider adopting placebo randomized controlled and blinded design to investigate long-term effects and optimal treatment protocols.

## Supporting information

**S1 File.  S1 File CONSORT.**
(DOCX)

**S2 File.  S2 File study protocol.**
(DOCX)

## Acknowledgments

The authors thank all participants.

## Author contributions

**Conceptualization:** Aysenur Karakus, Gokcen Akyurek.

**Formal analysis:** Aysenur Karakus, Esra Uzelpasaci, Gokcen Akyurek.

**Investigation:** Aysenur Karakus.

**Methodology:** Esra Uzelpasaci, Gokcen Akyurek.

**Writing – original draft:** Aysenur Karakus, Esra Uzelpasaci, Gokcen Akyurek.

**Writing – review & editing:** Aysenur Karakus, Esra Uzelpasaci, Gokcen Akyurek.

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
