## [Decision Letter · Decision Letter 0]

31 Oct 2024

PONE-D-24-36994The Effect of Progressive Relaxation Training on Pain Characteristics, Attack Frequency, Activity Self-Efficacy, and Pain-Related Disability in Women with Episodic Tension-Type Headache and Migraine: A Cohort StudyPLOS ONE

Dear Dr. KARAKUS,

Thank you for submitting your manuscript to PLOS ONE. After careful consideration, we feel that it has merit but does not fully meet PLOS ONE’s publication criteria as it currently stands. Therefore, we invite you to submit a revised version of the manuscript that addresses the points raised during the review process.

We look forward to receiving your revised manuscript.

Kind regards,

Ming Liu

Academic Editor

PLOS ONE

2. At PRTC, please ask the authors to include an explanation for the retrospective CT registration and confirmation that all related CTs are registered, using send back in ITC desk notes. At RTC, please check the authors' response and ping me if the authors do not address this.

3. In the online submission form, you indicated that [Data available on request from the author (A.K.)]. All PLOS journals now require all data underlying the findings described in their manuscript to be freely available to other researchers, either 1. In a public repository, 2. Within the manuscript itself, or 3. Uploaded as supplementary information. This policy applies to all data except where public deposition would breach compliance with the protocol approved by your research ethics board. If your data cannot be made publicly available for ethical or legal reasons (e.g., public availability would compromise patient privacy), please explain your reasons on resubmission and your exemption request will be escalated for approval.

Additional Editor Comments :

There are some flaws in this paper. The authors should revise the paper according to the reviewers' comments carefully.

Reviewers' comments:

Reviewer's Responses to Questions

**Comments to the Author**

1. Is the manuscript technically sound, and do the data support the conclusions?

Reviewer #1: No

Reviewer #2: Yes

2. Has the statistical analysis been performed appropriately and rigorously? 

Reviewer #1: No

Reviewer #2: Yes

3. Have the authors made all data underlying the findings in their manuscript fully available?

Reviewer #1: Yes

Reviewer #2: Yes

4. Is the manuscript presented in an intelligible fashion and written in standard English?

Reviewer #1: Yes

Reviewer #2: Yes

5. Review Comments to the Author

Reviewer #1: This is a cohort study, assessing the effect of PRT on various outcomes in women who suffer from Episodic headache tension and migraines. These have been considered as two separate groups.

In the introduction - can the authors state what the hypothesis is. I.e do you expect the PRT to be more effective in one group compared to the other? Or the hypothesis is that PRT is effective regardless of type of headaches.

The main concern, is the authors do not give enough information about the design of the study. Two groups are potentially receiving the same intervention and the sample size calculated is based on detecting group differences.

In a conventional RCT, the power calculation would be aimed at measuring the effect of the intervention compared with usual care. In this case its comparing between groups based on type disease which makes the design complex.

As the current study design suggest you have two groups presenting with different headache types but receiving the same intervention- should the question be looking at within difference instead of between? Since the between differences do not really make sense and make interpretation difficult - Can the authors comment on this point.

Line 122- Add the word This is a “Prospective” cohort study- as its clear that you collected data prospectively.

Inclusion criteria, include women with diagnosed THH - did the authors consider whether this would include newly diagnosed as well as those women who have suffered with this condition for a long period of time.

Unsure why authors have used the CONSORT checklist as this is not an RCT

Can all descriptive statistics in the table include median summaries as well, since n=20 in each group is a small sample.

Can the authors comment on the uneven distribution of education level between groups?

The analysis included individuals who adhered to the intervention - can this be stated in the methods section. A note that there would be some impact in the interpretation, i.e is this generalisable for women who do not adhere to the PRT? Perhaps this reflects the real world i.e would this indicate people affected by the condition and able to undertake the PRT?

Sample size calculation/information written in protocol is different to what is written in the paper - can this be made consistent.

Sample size planned was 40, but flow diagram shown 44 and conveniently 4 excluded women to non-adherence - this looks like biased???

Reviewer #2: Title: The Effect of Progressive Relaxation Training on Pain Characteristics, Attack Frequency, Activity Self-Efficacy, and Pain-Related Disability in Women with Episodic Tension-Type Headache and Migraine: A Cohort Study

Positive Aspects:

Relevant Topic: The study addresses a highly relevant issue in public health and clinical practice. Primary headaches, such as migraines and episodic tension-type headaches (TTH), are highly prevalent, especially affecting young women. Investigating the effectiveness of progressive relaxation training (PRT) in improving key variables like pain intensity, attack frequency, and pain-related disability is important to promote non-pharmacological interventions.

Appropriate Study Design: The manuscript presents a well-structured cohort design, with clearly defined outcome measures, such as the Visual Analog Scale (VAS), the Headache Impact Test (HIT-6), and the Pain Catastrophizing Scale (PCS). The PRT intervention was standardized, and the use of a comparison between migraine and tension-type headache groups strengthens the comparative analysis.

Coherent and Relevant Results: The results show a significant reduction in the intensity and frequency of pain attacks in both groups, with notable improvements in perceived activity self-efficacy and pain-coping abilities. This demonstrates the clinical utility of PRT for populations suffering from headaches, which could encourage its implementation in routine treatments.

Areas for Improvement:

Lack of Blinding: A significant limitation of the study is the absence of blinding, as the researchers administering the intervention also conducted the assessments. This introduces potential bias, especially in studies involving subjective interventions such as PRT. It is suggested that future research includes blinded evaluators.

Absence of a Placebo-Controlled Group: Although the study includes comparison groups (migraine and TTH), the lack of a placebo-controlled group limits the interpretation of results. A placebo group would have provided a more robust evaluation of the isolated effect of PRT on outcomes, reducing participant expectation bias.

Generalization of Results: The study focused exclusively on young women with episodic headaches, limiting generalization to other populations, such as men or patients with chronic headaches. Future studies should broaden the sample to include a more diverse population and explore whether the effects of PRT are similar in other age groups and headache types.

Detailing of Statistical Analysis Protocol: While the use of SPSS software was mentioned, it would be beneficial to include more details regarding the statistical tests applied, particularly in terms of multivariate analysis, corrections for multiple comparisons, and justifications for the sample size used.

Suggestions for Improvement:

Include a more robust section on the safety of the intervention, as muscle relaxation might impact musculoskeletal conditions.

Explore in greater depth the physiological differences between migraine and TTH patients and how these differences relate to the study’s findings.

6. PLOS authors have the option to publish the peer review history of their article (what does this mean? ). If published, this will include your full peer review and any attached files.

**Do you want your identity to be public for this peer review?** For information about this choice, including consent withdrawal, please see our Privacy Policy .

Reviewer #1: No

Reviewer #2: No

---

## [Author Response · Author response to Decision Letter 1]

5 Dec 2024

Response to Reviewers

Reviewer #1: This is a cohort study, assessing the effect of PRT on various outcomes in women who suffer from Episodic headache tension and migraines. These have been considered as two separate groups.

Dear Reviewer,

Thank you very much for your time and valuable feedback. The changes have been made and are presented in the Revised Manuscript with Track Changes File for your review.

Kind regards.

In the introduction - can the authors state what the hypothesis is. I.e do you expect the PRT to be more effective in one group compared to the other? Or the hypothesis is that PRT is effective regardless of type of headaches.

Response: Thank you for pointing this out. We hypothesized that PRT is expected to be more effective in one group compared to the other. The hypothesis statements have been revised to improve clarity in the manuscript. Based on your suggestions, we made some addition between lines 112-119. We hope that these changes will meet your expectations. However, if you have a further suggestion, we will revise the manuscript in accordance with your suggestion.

Added Text in Introduction: "The aim of the present study was to investigate comparative effectiveness of PRT on pain characteristics, attack frequency, activity self-efficacy, and pain related disability among women experiencing two different headache types, TTH and migraine.

The null hypothesis is that PRT has no differential effect on pain characteristics, attack frequency, activity self-efficacy, or pain-related disability between women with episodic TTH and those with migraine.

The alternative hypothesis is that PRT is more effective in one group (TTH or migraine) compared to the other in improving pain characteristics, attack frequency, activity self-efficacy, and pain-related disability."

The main concern, is the authors do not give enough information about the design of the study. Two groups are potentially receiving the same intervention and the sample size calculated is based on detecting group differences.

Response: Thank you for your comments regarding the study design and sample size calculation. Fichtel and Larsson (2001) conducted a similar study in adolescents with TTH and migraines, applying PRT to two different headache groups (R1). Therefore, in the Study Design section, we have added further clarifications to improve understanding of our design between lines 128-131. Additionally, We also rearranged the title to "The Comparative Effectiveness of Progressive Relaxation Training on Pain Characteristics, Attack Frequency, Activity Self-Efficacy, and Pain-Related Disability in Women with Episodic Tension-Type Headache and Migraine" to match the changes we made to the method. Furthermore, we have revised thsample size (between lines 223-231) in the statistical analysis section for clarity. We hope that these changes will meet your expectations. These changes have also been reflected in the abstract to ensure consistency.

However, if you have a further suggestion, we will revise the manuscript in accordance with your suggestion.

Added Text in Study Design Section: "This study is prospective, experimental between-group study with premeasurements and postmeasurements. In this study, TTH and migraines participated in PRT sessions twice a week for six weeks. The aim of this study was to evaluate and compare the outcomes of these two distinct groups after the training. "

Added text in Statistical analysis Section: The G* Power software (G* Power, Version 3.1.9.7 Franz Faul, Universität Kiel, Germany) was used to determine the sample size. The preliminary hypothesis was defined as difference in VAS scores over time (pre-test and post-test) and between groups. Accordingly, it was assumed that the time and group effects would have a moderate effect size in our study. With a Type I error rate of 𝛼=0.05 α=0.05 (95% confidence level) and a desired power of 1−𝛽=0.80 1−β=0.80, the required sample size for statistical analyses was calculated as 20 (R2).

Post-hoc power analysis was conducted after the study to evaluate whether the sample size used was sufficient. The post-hoc power analysis revealed that the power of the study was 98%. This indicates that the sample size used in the study was statistically sufficient

R1: Fichtel Å, Larsson B. Does relaxation treatment have differential effects on migraine and tension‐type headache in adolescents?. Headache. 2001;41(3): 290-296. doi: 10.1046/j.1526-4610.2001.111006290.x

R2: Faul F, Erdfelder E, Buchner A, Lang AG. 2009. Statistical power analyses using G* Power 3.1: Tests for correlation and regression analyses. Behavior Res Met, 41: 1149-1160)

In a conventional RCT, the power calculation would be aimed at measuring the effect of the intervention compared with usual care. In this case its comparing between groups based on type disease which makes the design complex.

Yes, you are right; we acknowledge that interpreting between-group differences could lead to complexity. However, the study aims to determine the effectiveness of PRT on headaches by using both within-group and between-group comparisons to evaluate its superior effect across different headache groups. Therefore, the study's power has been calculated based on the post-treatment between-group difference. To enhance clarity, sample size (between lines 223-228) and the power calculation has been revised (between lines 229-231).Thank you for your feedback.

Added text in Statistical analysis Section: The G* Power software (G* Power, Version 3.1.9.7 Franz Faul, Universität Kiel, Germany) was used to determine the sample size. The preliminary hypothesis was defined as difference in VAS scores over time (pre-test and post-test) and between groups. Accordingly, it was assumed that the time and group effects would have a moderate effect size in our study. With a Type I error rate of 𝛼=0.05 α=0.05 (95% confidence level) and a desired power of 1−𝛽=0.80 1−β=0.80, the required sample size for statistical analyses was calculated as 20. Post-hoc power analysis was conducted after the study to evaluate whether the sample size used was sufficient (R1).

The post-hoc power analysis revealed that the power of the study was 98%. This indicates that the sample size used in the study was statistically sufficient

R1: Faul F, Erdfelder E, Buchner A, Lang AG. 2009. Statistical power analyses using G* Power 3.1: Tests for correlation and regression analyses. Behavior Res Met, 41: 1149-1160)

As the current study design suggest you have two groups presenting with different headache types but receiving the same intervention- should the question be looking at within difference instead of between? Since the between differences do not really make sense and make interpretation difficult - Can the authors comment on this point.

Yes, you are right. Thank you for this valuable insight. We believe that focusing on within group differences could provide a clearer perspective on the effectiveness of PRT for each headache type individually. However, our study aims to examine not only the treatment effect within each group but also to explore whether PRT may exhibit a differential impact between the two headache types. By assessing both within-group and between-group differences, we aim to identify if one group may experience a more pronounced benefit from PRT compared to the other. In our study, within-group comparisons revealed that PRT was effective in both headache groups. Moreover, between-group comparisons indicated that its effect was higher in the TTH group. In this context, delta scores have been added to the result section to show the difference between the groups to increase clarity (lines 302-330). Also, we made updates to the discussion section to enhance clarity and comprehensibility (between lines 340-348). We hope that these changes will meet your expectations. However, if you have a further suggestion, we will revise the manuscript in accordance with your suggestion.

Added Text in Result Section: Table 4 and its interpretation added in between in the lines 302 -330.

Added text in Discussion Section: " This study evaluated the comparative effectiveness of PRT in two different types of headaches, TTH and migraine, by assessing its effects on pain characteristics, attack frequency, activity self-efficacy, and pain-related disability in women. Both within-group and between-group differences were analyzed to determine whether one group experienced more significant benefits from PRT compared to the other. The results revealed that PRT was effective in reducing pain intensity and attack frequency, while also positively impacting activity self-efficacy and pain-related disability in women with both TTH and migraine. Notably, between-group comparisons demonstrated that PRT was more effective in the TTH group. "

Line 122- Add the word This is a “Prospective” cohort study- as its clear that you collected data prospectively.

Response: Thank you for pointing this out. Line 128 – We have revised the sentence to better reflect the study design.

Added Text in Study Design Section: This study is prospective, experimental between-group study with premeasurements and postmeasurements.

Inclusion criteria, include women with diagnosed THH - did the authors consider whether this would include newly diagnosed as well as those women who have suffered with this condition for a long period of time.

Response: Since women with migraines are more likely to use migraine prophylaxis or pain relievers, we included only newly diagnosed women or those who had experienced headaches for at least 3 months at their initial consultation with a neurology specialist. This criterion helps minimize the influence of previous treatments on our study outcomes. For women with tension-type headaches (TTH), where the pain is generally more tolerable and often managed without immediate pharmacological intervention, we included those who had been experiencing headaches for a duration of 3 to 6 months (between lines 142-143). These adjustments have been added to the inclusion criteria section to enhance the study's focus on newly diagnosed cases and reduce confounding factors (R2, R3, R4).

We hope that these changes will meet your expectations. However, if you have a further suggestion, we will revise the manuscript in accordance with your suggestion.

Added Text in Participants Section: "Women diagnosed with migraines within the last 3 months and women diagnosed with TTH within the last 3 to 6 months were included in the study".

R2. Diener HC, Holle D, Dresler T, Gaul C. Chronic Headache Due to Overuse of Analgesics and Anti-Migraine Agents. Dtsch Arztebl Int. 2018 Jun 1;115(22):365-370. doi: 10.3238/arztebl.2018.0365. PMID: 29932046; PMCID: PMC6039717.

R3. Söderberg E, Carlsson J, Stener-Victorin E. Chronic Tension-Type Headache Treated with Acupuncture, Physical Training and Relaxation Training. Between-Group Differences. Cephalalgia. 2006;26(11):1320-1329. doi:10.1111/j.1468-2982.2006.01209.x

R4. Headache Classification Committee of the International Headache Society (IHS) The International Classification of Headache Disorders, 3rd edition. Cephalalgia. 2018;38(1):1-211. doi: 10.1177/0333102417738202 https://doi.org/10.1177/0333102417738202

Unsure why authors have used the CONSORT checklist as this is not an RCT

Response: Yes, you are right. Thank you for highlighting this. Initially, we uploaded the STROBE checklist as our primary checklist. At the request of the technical editor, we added the CONSORT checklist when submitting our article. However, as a result, both the STROBE and CONSORT checklists were present in the file folder.

Can all descriptive statistics in the table include median summaries as well, since n=20 in each group is a small sample.

Response: Thank you for your feedback. Given the small sample size of n=20 in each group, we have included median summaries in addition to the other descriptive statistics in the table to provide a more robust representation of the data.

Tables have been updated in the Results section.

Can the authors comment on the uneven distribution of education level between groups?

Thank you for pointing this out. In our study, women with higher education levels were present in both groups. 95.0% of women in the Migraine group were university graduates, while in the TTH group, 55.0% were university graduates. In the literature, the results regarding the relationship between education level and TTH and migraine are controversial. Some studies associate high education level with migraine and TTH (R5, R6), while in others, the effect of education level is modest (R7,R8,R9). Education reflects an individual's intellectual abilities and may also be associated with different health behaviors (R10). In the study, women had a higher level of education. This may have provided an advantage in seeing the actual effect of the training given to both groups. This point is included in the strengths of the study (between lines 440-446)

Added Text in the Discussion-Strengths of the Study: ‘’ In the study, women with higher education levels were present in both groups. 95.0% of women in the Migraine group were university graduates, while in the TTH group, 55.0% were university graduates. Education reflects an individual's intellectual abilities and may also be associated with different health behaviors (47). Both women with TTH and migraine were a high level of education. It may have been advantageous in enabling patients to better understand and apply the PRT and thus reveal the true effect of the PRT in both groups (47).’’

R5. Yang, H., Pu, S., Yang, L., Luo, W., Zhao, J., Liu, E., … & Luo, J. (2022). Migraine among students of a medical college in western china: a cross-sectional study. European Journal of Medical Research, 27(1). https://doi.org/10.1186/s40001-022-00698-9

R6. Vuković, V., Plavec, D., Pavelin, S., Jančuljak, D., Ivanković, M., & Demarin, V. (2010). Prevalence of migraine, probable migraine and tension-type headache in the croatian population. Neuroepidemiology, 35(1), 59-65. https://doi.org/10.1159/000310940

R7. Ertaş, M., Baykan, B., Orhan, E., Zarifoğlu, M., Karlı, N., Saıp, S., … & Síva, A. (2012). One-year prevalence and the impact of migraine and tension-type headache in turkey: a nationwide home-based study in adults. The Journal of Headache and Pain, 13(2), 147-157. https://doi.org/10.1007/s10194-011-0414-5

R8. Molarius A, Tegelberg A, Ohrvik J. Socio-economic factors, lifestyle, and headache disorders - a population-based study in Sweden. Headache. 2008 Nov-Dec;48(10):1426-37. doi: 10.1111/j.1526-4610.2008.01178.x. Epub 2008 Jul 1. PMID: 18624712.

R9. Han, L., Tfelt‐Hansen, P., Skytthe, A., Kyvik, K., & Olesen, J. (2011). Association between migraine, lifestyle and socioeconomic factors: a population-based cross-sectional study. The Journal of Headache and Pain, 12(2), 157-172. https://doi.org/10.1007/s10194-011-0321-9

R10. Skalamera, J. and Hummer, R. (2016). Educational attainment and the clustering of health-related behavior among u.s. young adults. Preventive Medicine, 84, 83-89. https://doi.org/10.1016/j.ypmed.2015.12.011

The analysis included individuals who adhered to the intervention - can this be stated in the methods section. A note that there would be some impact in the interpretation, i.e is this generalisable for women who do not adhere to the PRT? Perhaps this reflects the real world i.e would this indicate people affected by the condition and able to undertake the PRT?

Yes, you are correct. Thank you for your valuable comment. In the flowchart, we inadvertently did not use the correct terms. In the study, the women did not drop out due to adverse effects of the treatment or inability to comply with it. Some participants moved to another city, while others were unable to complete the final assessments or maintain the desired level of continuity in the study due to family matters or changes in their health conditions. Therefore, the study's flowchart has been revised accordingly.

Sample size calculation/information written in protocol is different to what is written in the paper - can this be made consistent.

Thank you for your comment. We made consistent sample size in Study Protocol and the Ma

---

## [Decision Letter · Decision Letter 1]

16 Jan 2025

PONE-D-24-36994R1The Comparative Effectiveness of Progressive Relaxation Training on Pain Characteristics, Attack Frequency, Activity Self-Efficacy, and Pain-Related Disability in Women with Episodic Tension-Type Headache and MigrainePLOS ONE

Dear Dr. KARAKUS,

Thank you for submitting your manuscript to PLOS ONE. After careful consideration, we feel that it has merit but does not fully meet PLOS ONE’s publication criteria as it currently stands. Therefore, we invite you to submit a revised version of the manuscript that addresses the points raised during the review process.

We look forward to receiving your revised manuscript.

Kind regards,

Ming Liu

Academic Editor

PLOS ONE

Journal Requirements:

Additional Editor Comments:

Please revise the paper carefully following the reviewers comments.

Reviewers' comments:

Reviewer's Responses to Questions

**Comments to the Author**

1. If the authors have adequately addressed your comments raised in a previous round of review and you feel that this manuscript is now acceptable for publication, you may indicate that here to bypass the “Comments to the Author” section, enter your conflict of interest statement in the “Confidential to Editor” section, and submit your "Accept" recommendation.

Reviewer #2: All comments have been addressed

Reviewer #3: All comments have been addressed

Reviewer #4: All comments have been addressed

Reviewer #5: (No Response)

2. Is the manuscript technically sound, and do the data support the conclusions?

Reviewer #2: Yes

Reviewer #3: Partly

Reviewer #4: Partly

Reviewer #5: Yes

3. Has the statistical analysis been performed appropriately and rigorously? 

Reviewer #2: Yes

Reviewer #3: Yes

Reviewer #4: Yes

Reviewer #5: Yes

4. Have the authors made all data underlying the findings in their manuscript fully available?

Reviewer #2: Yes

Reviewer #3: Yes

Reviewer #4: Yes

Reviewer #5: Yes

5. Is the manuscript presented in an intelligible fashion and written in standard English?

Reviewer #2: Yes

Reviewer #3: No

Reviewer #4: No

Reviewer #5: Yes

6. Review Comments to the Author

Reviewer #2: (No Response)

Reviewer #3: (No Response)

Reviewer #4: Sincerely

The study was statistically and methodologically reviewed.

Please make the following corrections:

The article is a clinical trial exploring the effectiveness of progressive relaxation training (PRT) in women with episodic tension-type headache (TTH) and migraine. Below are the identified flaws and areas for improvement:

The lack of evaluator blinding introduces bias, especially in subjective assessments such as pain scales.

The study does not include a placebo or alternative intervention group, which limits the robustness of the conclusions regarding the specific effects of PRT.

While a post-hoc analysis showed sufficient power, the small sample size (n=20 per group) may affect the generalizability of results and could lead to an overestimation of effect sizes.

Median summaries are not consistently provided despite the small sample size, which might skew the interpretation of results.

The paper does not mention adjustments for multiple statistical tests, potentially inflating the risk of Type I errors.

Limited details about multivariate analysis or rationale for specific tests reduce the clarity of the statistical approach.

Inclusion of only women within a narrow age range (20–45 years) limits the applicability of findings to broader populations, including men and older individuals.

The inclusion criteria (recent diagnoses for migraines and TTH) might not reflect chronic headache sufferers, a significant subgroup in clinical practice.

The rationale for comparing outcomes between two different headache types, which inherently have different pathophysiological mechanisms, is unclear and complicates interpretation.

The discussion lacks depth in explaining how PRT impacts the distinct mechanisms of migraine and TTH.

Uneven distribution of education levels between groups is noted but not adequately discussed regarding its potential impact on outcomes.

Although ethical approval is mentioned, additional details about how informed consent was obtained and adherence monitoring are absent.

The exclusion of non-adherent participants raises questions about the intervention's feasibility and applicability in real-world settings.

While safety measures are briefly described, a dedicated section detailing adverse effects or participant concerns during PRT sessions is missing.

The inclusion of a CONSORT checklist for a non-RCT study is inappropriate and inconsistent.

Incorporate a placebo or active comparator group in future studies.

Ensure evaluator blinding to reduce bias.

Expand the sample to include diverse demographic groups for better generalizability.

Provide detailed explanations of statistical methods, particularly corrections for multiple testing.

Address the uneven distribution of education levels and its implications for the findings.

Include a robust discussion of PRT's mechanisms of action on migraine and TTH.

Offer a clearer rationale for between-group comparisons and their relevance to clinical practice.

Good luck.

Reviewer #5: Good paper with enough information, but I did not find all details related to consort checklist. The method section has lots of missing points: like sample size, two group explanation, randomization and blinding process.

7. PLOS authors have the option to publish the peer review history of their article (what does this mean? ). If published, this will include your full peer review and any attached files.

**Do you want your identity to be public for this peer review?** For information about this choice, including consent withdrawal, please see our Privacy Policy .

Reviewer #2: No

Reviewer #3: **Yes: ** Dr Aman Suresh T Assistant Professor , Faculty of Pharmacy Dr MGR educational trust and research institute

Reviewer #4: **Yes: ** Ebrahim Abbasi

Reviewer #5: No

---

## [Author Response · Author response to Decision Letter 2]

24 Jan 2025

PONE-D-24-36994R1

The Comparative Effectiveness of Progressive Relaxation Training on Pain Characteristics, Attack Frequency, Activity Self-Efficacy, and Pain-Related Disability in Women with Episodic Tension-Type Headache and Migraine

PLOS ONE

Comments to the Author

5. Is the manuscript presented in an intelligible fashion and written in standard English?

Reviewer #2: Yes

Reviewer #3: No

Reviewer #4: No

Reviewer #5: Yes

Thank you for your comment. The manuscript has been thoroughly reviewed and revised to ensure it is presented in an intelligible fashion and written in standard English.

The study was statistically and methodologically reviewed.

Please make the following corrections:

The article is a clinical trial exploring the effectiveness of progressive relaxation training (PRT) in women with episodic tension-type headache (TTH) and migraine. Below are the identified flaws and areas for improvement:

The lack of evaluator blinding introduces bias, especially in subjective assessments such as pain scales.

Thank you for your comment. Yes, you are right. To minimize bias in the study, the statistician was blinded. In the literature, most studies conducted on conditions such as migraine and tension-type headaches have used the VAS (Visual Analogue Scale) (R1, R2). Therefore, we also used a subjective assessment method and women were asked to mark the intensity of their headaches on the horizontal lines and the results were recorded in centimeters. Additionally, the evaluators ensured that the women completed the forms entirely based on self-reported pain scales. However, the lack of blinding for the evaluators and participants regarding the study group represents a limitation. This has been addressed in the limitations section of the manuscript.

R1: Peng, K. P., & Wang, S. J. (2012). Migraine diagnosis: screening items, instruments, and scales. Acta Anaesthesiologica Taiwanica, 50(2), 69-73.

R2: Houts, C. R., McGinley, J. S., Nishida, T. K., Buse, D. C., Wirth, R. J., Dodick, D. W., ... & Lipton, R. B. (2021). Systematic review of outcomes and endpoints in acute migraine clinical trials. Headache: The Journal of Head and Face Pain, 61(2), 263-275.

The study does not include a placebo or alternative intervention group, which limits the robustness of the conclusions regarding the specific effects of PRT.

Yes, you are right. In the study, most of the women with migraine and tension-type headache tend to start medication use in the early stages. The literature also indicates that medication use begins early in cases of severe pain (R1,R2). For this reason, obtaining the desired control group was challenging. Nevertheless, the absence of placebo and control groups has been included as a limitation.

R1: Bigal, M. E., & Lipton, R. B. (2009). Overuse of acute migraine medications and migraine chronification. Current pain and headache reports, 13, 301-307.

R2: Zwart, J. A., Dyb, G., Hagen, K. M. D. P., Svebak, S., & Holmen, J. M. D. P. (2003). Analgesic use: a predictor of chronic pain and medication overuse headache: the head–HUNT study. Neurology, 61(2), 160-164.

While a post-hoc analysis showed sufficient power, the small sample size (n=20 per group) may affect the generalizability of results and could lead to an overestimation of effect sizes. Median summaries are not consistently provided despite the small sample size, which might skew the interpretation of results.

Yes you are right. Post-hoc power analysis was conducted after the study to evaluate whether the sample size used was sufficient. The post-hoc power analysis revealed that the power of the study was 98%. This indicates that the sample size used in the study was statistically sufficient. Therefore, the Statistical Analysis section has been revised to include additional details (between lines 229-237). Median scores and delta scores have been expanded by adding Table (1),(2),(3) and (4) (between lines 302-330).Thank you for your feedback.

Added text in Statistical analysis Section: The G* Power software (G* Power, Version 3.1.9.7 Franz Faul, Universität Kiel, Germany) was used to determine the sample size. The preliminary hypothesis was defined as difference in VAS scores over time (pre-test and post-test) and between groups. Accordingly, it was assumed that the time and group effects would have a moderate effect size in our study. With a Type I error rate of 𝛼=0.05 α=0.05 (95% confidence level) and a desired power of 1−𝛽=0.80 1−β=0.80, the required sample size for statistical analyses was calculated as 20 (n=20) (R1).

Post-hoc power analysis was conducted after the study to evaluate whether the sample size used was sufficient.

The post-hoc power analysis revealed that the power of the study was 98%. This indicates that the sample size used in the study was statistically sufficient.

R1: Faul F, Erdfelder E, Buchner A, Lang AG. 2009. Statistical power analyses using G* Power 3.1: Tests for correlation and regression analyses. Behavior Res Met, 41: 1149-1160)

The paper does not mention adjustments for multiple statistical tests, potentially inflating the risk of Type I errors. Limited details about multivariate analysis or rationale for specific tests reduce the clarity of the statistical approach.

Thank your for comment. The Statistical Analysis section has been revised to include additional details (between lines 308-237). Median scores and delta scores have been expanded by adding Table 4 (between lines 302-330).

Added text in Statistical analysis Section: The G* Power software (G* Power, Version 3.1.9.7 Franz Faul, Universität Kiel, Germany) was used to determine the sample size. The preliminary hypothesis was defined as difference in VAS scores over time (pre-test and post-test) and between groups. Accordingly, it was assumed that the time and group effects would have a moderate effect size in our study. With a Type I error rate of 𝛼=0.05 α=0.05 (95% confidence level) and a desired power of 1−𝛽=0.80 1−β=0.80, the required sample size for statistical analyses was calculated as 20 (n=20) (R1).. If you have any other corrections or modifications you would like to request, we are open to making them.

R1: Faul F, Erdfelder E, Buchner A, Lang AG. 2009. Statistical power analyses using G* Power 3.1: Tests for correlation and regression analyses. Behavior Res Met, 41: 1149-1160)

Inclusion of only women within a narrow age range (20–45 years) limits the applicability of findings to broader populations, including men and older individuals. The inclusion criteria (recent diagnoses for migraines and TTH) might not reflect chronic headache sufferers, a significant subgroup in clinical practice.

Thank you for your valuable feedback. As mentioned in the literature, migraine and TTH are predominantly seen in women wtihin narrow age range, which guided our focus on this population (R3,R4). Additionally, individuals with chronic migraine are likely to have more established health-seeking behaviors and medication use compared to those with episodic migraine (R5,R6). These factors may influence both their treatment outcomes and their participation in non-pharmacological interventions like Progressive Relaxation Training . While our study specifically targeted women with episodic headaches, we agree that future research should explore the effects of PRT in individuals with chronic migraine and TTH, as well as in broader and more diverse populations, to enhance the generalizability of the findings. We have included this in the discussion and recommendations for future studies. However, if you have a further suggestion, we will revise the manuscript in accordance with your suggestion.

R3: Smitherman, T. A., Burch, R., Sheikh, H., & Loder, E. (2013). The Prevalence, Impact, and Treatment of Migraine and Severe Headaches in the U nited S tates: A Review of Statistics From National Surveillance Studies. Headache: The Journal of Head and Face Pain, 53(3), 427-436.

R4:Neumeier, M. S., Pohl, H., Sandor, P. S., Gut, H., Merki-Feld, G. S., & Andrée, C. (2021). Dealing with headache: sex differences in the burden of migraine-and tension-type headache. Brain sciences, 11(10), 1323.

R5: Seng, E. K., Martin, P. R., & Houle, T. T. (2022). Lifestyle factors and migraine. The Lancet Neurology, 21(10), 911-921.

R6: Hagen, K., Åsberg, A. N., Stovner, L., Linde, M., Zwart, J. A., Winsvold, B. S., & Heuch, I. (2018). Lifestyle factors and risk of migraine and tension-type headache. Follow-up data from the Nord-Trøndelag Health Surveys 1995–1997 and 2006–2008. Cephalalgia, 38(13), 1919-1926.

The rationale for comparing outcomes between two different headache types, which inherently have different pathophysiological mechanisms, is unclear and complicates interpretation.

Response: Thank you for your comments regarding the study design and different pathophysiological mechanisms of headaches. Fichtel and Larsson (2001) conducted a similar study in adolescents with TTH and migraines, applying PRT to two different headache groups (R1). Therefore, in the Study Design section, we have added further clarifications to improve understanding of our design between lines 128-131. Additionally, We also rearranged the title to "The Comparative Effectiveness of Progressive Relaxation Training on Pain Characteristics, Attack Frequency, Activity Self-Efficacy, and Pain-Related Disability in Women with Episodic Tension-Type Headache and Migraine" to match the changes we made to the method. Furthermore, we have revised the sample size (between lines 229-231) in the statistical analysis section for clarity. We hope that these changes will meet your expectations. These changes have also been reflected in the abstract to ensure consistency.

However, if you have a further suggestion, we will revise the manuscript in accordance with your suggestion.

Added Text in Study Design Section: "This study is prospective, experimental between-group study with premeasurements and postmeasurements. In this study, TTH and migraines participated in PRT sessions twice a week for six weeks. The aim of this study was to evaluate and compare the outcomes of these two distinct groups after the training.

The discussion lacks depth in explaining how PRT impacts the distinct mechanisms of migraine and TTH.

Thank you for your comment. The details regarding the pathophysiology of Migraine and TTH, as well as the Progressive Relaxation Training, are provided between lines 382 and 399 in the manuscript. However, we have expanded the discussion to explore the physiological differences between migraine and tension-type headache (TTH) patients. Migraine is often associated with neurovascular mechanisms and heightened central sensitization, whereas TTH is predominantly linked to muscular tension and peripheral pain mechanisms [32,33]. These differences may influence the effectiveness of Progressive Relaxation Training, which targets muscle tension and promotes overall relaxation. Our findings suggest that while both groups benefit from PRT, the mechanisms underlying these benefits may differ. This has been discussed in the manuscript to provide a more comprehensive understanding of how these physiological differences relate to our study’s discussion (between lines 378-381). We hope that these changes will meet your expectations. However, if you have a further suggestion, we will revise the manuscript in accordance with your suggestion.

Added text in Discussion Section: Migraine is often associated with neurovascular mechanisms and heightened central sensitization, whereas TTH is predominantly linked to muscular tension and peripheral pain mechanisms. These differences may influence the effectiveness of PRT, which targets muscle tension and promotes overall relaxation [4,32,33].

Thank you very much for your comments. Revisions have been made based on your suggestions. We are open to making further adjustments if you require.

Uneven distribution of education levels between groups is noted but not adequately discussed regarding its potential impact on outcomes.

Thank you for pointing this out. In our study, women with higher education levels were present in both groups. 95.0% of women in the Migraine group were university graduates, while in the TTH group, 55.0% were university graduates. In the literature, the results regarding the relationship between education level and TTH and migraine are controversial. Some studies associate high education level with migraine and TTH (R5, R6), while in others, the effect of education level is modest (R7,R8,R9). Education reflects an individual's intellectual abilities and may also be associated with different health behaviors (R10). In the study, women had a higher level of education. This may have provided an advantage in seeing the actual effect of the training given to both groups. This point is included in the strengths of the study. However, if you have a further suggestion, we will revise the manuscript in accordance with your suggestion.

Added Text in the Discussion-Strengths of the Study:‘’In the study, women with higher education levels were present in both groups. 95.0% of women in the Migraine group were university graduates, while in the TTH group, 55.0% were university graduates. Education reflects an individual's intellectual abilities and may also be associated with different health behaviors (47). Both women with TTH and migraine were a high level of education. It may have been advantageous in enabling patients to better understand and apply the PRT and thus reveal the true effect of the PRT in both groups (47).

R5. Yang, H., Pu, S., Yang, L., Luo, W., Zhao, J., Liu, E., … & Luo, J. (2022). Migraine among students of a medical college in western china: a cross-sectional study. European Journal of Medical Research, 27(1). https://doi.org/10.1186/s40001-022-00698-9

R6. Vuković, V., Plavec, D., Pavelin, S., Jančuljak, D., Ivanković, M., & Demarin, V. (2010). Prevalence of migraine, probable migraine and tension-type headache in the croatian population. Neuroepidemiology, 35(1), 59-65. https://doi.org/10.1159/000310940

R7. Ertaş, M., Baykan, B., Orhan, E., Zarifoğlu, M., Karlı, N., Saıp, S., … & Síva, A. (2012). One-year prevalence and the impact of migraine and tension-type headache in turkey: a nationwide home-based study in adults. The Journal of Headache and Pain, 13(2), 147-157. https://doi.org/10.1007/s10194-011-0414-5

R8. Molarius A, Tegelberg A, Ohrvik J. Socio-economic factors, lifestyle, and headache disorders - a population-based study in Sweden. Headache. 2008 Nov-Dec;48(10):1426-37. doi: 10.1111/j.1526-4610.2008.01178.x. Epub 2008 Jul 1. PMID: 18624712.

R9. Han, L., Tfelt‐Hansen, P., Skytthe, A., Kyvik, K., & Olesen, J. (2011). Association between migraine, lifestyle and socioeconomic factors: a population-based cross-sectional study. The Journal of Headache and Pain, 12(2), 157-172. https://doi.org/10.1007/s10194-011-0321-9

R10. Skalamera, J. and Hummer, R. (2016). Educational attainment and the clustering of health-related behavior among u.s. young adults. Preventive Medicine, 84, 83-89. https://doi.org/10.1016/j.ypmed.2015.12.011

Although ethical approval is mentioned, additional details about how informed consent was obtained and adherence monitoring are absent.

Thank you for your comments regarding the ethical considerations of our study. We have added a section to the manuscript that outlines the process through which informed consent was secured from all participants. This includes details on the information provided to participants regarding the nature and purpose of the study, potential risks and benefits, and their rights to withdraw at any time without any consequences. Furthermore, we have included additional information on how we monitored adherence to the Progressive Relaxation Training (PRT) throughout the study. This monitoring was conducted through weekly meetings and regular phone calls with the participants. These methods were applied to ensure continuous participation and to add

---

## [Decision Letter · Decision Letter 2]

21 Feb 2025

The Comparative Effectiveness of Progressive Relaxation Training on Pain Characteristics, Attack Frequency, Activity Self-Efficacy, and Pain-Related Disability in Women with Episodic Tension-Type Headache and Migraine

PONE-D-24-36994R2

Dear Dr. KARAKUS,

We’re pleased to inform you that your manuscript has been judged scientifically suitable for publication and will be formally accepted for publication once it meets all outstanding technical requirements.

Kind regards,

Ming Liu

Academic Editor

PLOS ONE

Additional Editor Comments (optional):

Reviewers' comments:

Reviewer's Responses to Questions

**Comments to the Author**

1. If the authors have adequately addressed your comments raised in a previous round of review and you feel that this manuscript is now acceptable for publication, you may indicate that here to bypass the “Comments to the Author” section, enter your conflict of interest statement in the “Confidential to Editor” section, and submit your "Accept" recommendation.

Reviewer #1: All comments have been addressed

Reviewer #6: All comments have been addressed

2. Is the manuscript technically sound, and do the data support the conclusions?

Reviewer #1: Yes

Reviewer #6: Yes

3. Has the statistical analysis been performed appropriately and rigorously? 

Reviewer #1: Yes

Reviewer #6: Yes

4. Have the authors made all data underlying the findings in their manuscript fully available?

Reviewer #1: Yes

Reviewer #6: Yes

5. Is the manuscript presented in an intelligible fashion and written in standard English?

Reviewer #1: Yes

Reviewer #6: Yes

6. Review Comments to the Author

Reviewer #1: All comments have been addressed

Reviewer #6: Dear author,

Thank you for your nice work. You already incorporated all my comment and suggestion . I suggested for publication and keep it up the nice work to feed the world community. Once again congratulations for the interesting work. This is what is what is expected from us and let we play our responsibility as professional.

7. PLOS authors have the option to publish the peer review history of their article (what does this mean? ). If published, this will include your full peer review and any attached files.

**Do you want your identity to be public for this peer review?** For information about this choice, including consent withdrawal, please see our Privacy Policy .

Reviewer #1: No

Reviewer #6: No

---

## [Editor Report · Acceptance letter]

PONE-D-24-36994R2

PLOS ONE

Dear Dr. Karakus,

I'm pleased to inform you that your manuscript has been deemed suitable for publication in PLOS ONE. Congratulations! Your manuscript is now being handed over to our production team.

Kind regards,

on behalf of

Professor Ming Liu

Academic Editor

PLOS ONE